

# A comparison of the effects of two protocols of concurrent resistance and aerobic training on physical fitness in middle school students

Zhen Li[1],[*], Teng Ding[2],[*], Yanan Gao[1], Xiaowei Han[3], Yang Liu[4] and Zhixiong Zhou[5]

[1] School of Physical Education and Sport Science, Fujian Normal University, Fuzhou, Fujian, China
[2] School of Physical Education, Ningxia University, Yinchuan, Ningxia, China
[3] Faculty of Education, Beijing Normal University, Beijing, Beijing, China
[4] Hebei Institute of International Business and Economics, Qinhuangdao, Hebei, China
[5] Institute of Artificial Intelligence in Sports, Capital University of Physical Education and Sports, Beijing, China
[*] These authors contributed equally to this work.

Corresponding author
Zhixiong Zhou,
zhouzhixiong@cupes.edu.cn

## ABSTRACT

**Objective:** This study aimed to compare the effects of two concurrent training (CT) protocols on the physical fitness of middle school students.

**Method:** A 12-week quasi-experimental pre-test/post-test study was conducted with 157 middle school students (age = $12.48 \pm 0.34$, $n = 90$ females) divided into three groups: CT group A (CT-0h) received combined resistance training (RT) and aerobic training (AT) in each physical education session, CT group B (CT-48h) received RT and AT across two separate physical education classes 48 h apart, and a control group (Con) received no training. Training occurred twice a week. Test indicators included cardiorespiratory fitness (CRF) measured by estimated $VO_2max$ and 20 m shuttle run (laps), as well as muscle strength assessed through long jump, vertical jump, and handgrip strength.

**Results:** The intervention groups exhibited significant increases in estimated $VO_2max$ and muscle strength compared to their baseline values ($p < 0.05$). Both CT-0h and CT-48h groups demonstrated significant improvements in 20 m shuttle run (laps) (mean difference: 8.88 laps, $p < 0.01$; mean difference: 4.81 laps, $p < 0.01$, respectively), standing long jump (mean difference: 6.20 cm, $p < 0.01$; mean difference: 3.68 cm, $p < 0.01$, respectively), vertical jump (mean difference: 4.95 cm, $p < 0.01$; mean difference: 4.04 cm, $p < 0.01$, respectively), and handgrip strength (mean difference: 11.17 kg, $p < 0.01$; mean difference: 6.99 kg, $p < 0.01$, respectively). CT-0h group exhibited significantly increased estimated $VO_2max$ (mean difference: 1.47 ml/kg/min, $p < 0.01$) compared to the CT-48h group.

**Conclusion:** Both CT programs effectively improved adolescents' physical fitness indicators. However, the program that integrated RT and AT within the same physical education class demonstrated superior enhancement in adolescents' CRF.

## INTRODUCTION

Declining levels of physical activity and increased sedentary behaviors have become widespread problems among adolescents worldwide (*Loo et al., 2022*; *Valanju et al., 2022*). As a direct result, problems such as being overweight, having reduced cardiorespiratory endurance, and cardiometabolic dysfunction are beginning to appear in the adolescent population. Without timely intervention, these problems significantly increase the risk of cardiovascular disease, type II diabetes, and cardiovascular disease in adulthood. Therefore, innovative solutions are urgently needed to address this public health challenge (*Duft et al., 2020*).

Evidence suggests that school-based interventions are effective at improving young people's physical health. For instance, increasing high-intensity physical activity is effective at improving child and adolescent cardiorespiratory endurance (*Cao et al., 2022*; *Hartwig et al., 2021*). School physical education classes offer opportunities for young people to engage in physical activity and health promotion (*Hartwig et al., 2021*). However, it has also been suggested that school physical education classes are ineffective at improving adolescent fitness, possibly because most school-based intervention strategies focus on enhancing adolescents' cardiorespiratory fitness (CRF) through aerobic training (AT) while neglecting muscular strength (*Alves et al., 2018*). There is considerable evidence that levels of muscle health in adolescence are highly correlated with future health, and there is an international consensus that musculoskeletal health assessments should be included in adolescent health assessments (*Janz et al., 2021*; *Moeskops et al., 2022*; *Stricker, Faigenbaum & McCambridge, 2020*). Additionally, research has shown that maintaining a high level of physical fitness throughout adolescence might predict daily living activities in adulthood and even in old age, as well as lower the risk of cognitive decline (*Liu, Liu & Li, 2021*).

Given the significance of physical fitness for adolescent health, international organizations like the World Health Organization (WHO) and the American College of Sports and Medicine (ACSM) have recommended AT and resistance training (RT) as part of physical activity guidelines for adolescents to support the development of their physical fitness (*De la Corte-Rodriguez et al., 2021*; *King et al., 2019*). Considering that most adolescents only engage in relevant physical activity in physical education classes, embedding AT and RT into the school environment and implementing safety training programs for adolescents to enhance and promote physical activity seems to be an effective approach (*Beets et al., 2016*).

The combination of AT and RT is generally defined as concurrent training (CT) (*Wilson et al., 2012*). CT builds strength and increases aerobic capacity, saves time and lessens training monotony, and has drawn a lot of interest from researchers in recent years (*Jha et al., 2018*; *Sousa et al., 2019*). However, researchers are currently primarily focused on the "interference effect" of the training schedule of CT on strength development in adults or athletes. For example, *Sousa et al. (2019)* concluded that the intensity schedule of CT should be arranged in such a way that higher RT loads are combined with lower AT intensities in order to increase strength gains and reduce post-training losses. It has also
been suggested that the interval between AT and RT is a major factor in the "interference effect"; *i.e.*, performing CT during the same training session (with no interval) or less than 6 h apart in a single day reduces muscle strength improvement and results in an interference effect, but alternating days (>24 h) between AT and RT does not (*Robineau et al., 2016*; *Schumann et al., 2022*; *Vechin et al., 2021*). Given the anthropometric, physiological, and biomechanical variations between adolescents and adults, it is crucial to stress that this theory is based on adult data, and it has remained unclear whether it can be applied to the adolescent population (*Jansson et al., 2022*).

Prior research has posited that the interference effect may be age-related. For instance, *Bluett, De Ste Croix & Lloyd (2015)* discovered that, in comparison to isolated AT, CT led to a decline in the 3 km running performance of adolescents aged 10 to 13. However, *Spurrs, Murphy & Watsford (2003)* arrived at a contrary conclusion based on adult data. Through a meta-analysis, *Gabler et al. (2018)* found that CT was more effective than single-mode AT or RT at enhancing the physical fitness and exercise performance of adolescents. Specifically, CT demonstrated improvements in exercise performance, particularly in children and adolescents, when compared to AT. Furthermore, CT was found to be more effective than RT at enhancing muscular strength in adolescents. Thus, it is evident that applying adult data to children and adolescents, in light of the physiological processes associated with growth and maturation, lacks validity.

Considering the diversity of CT programs and the minimal hormonal impact of exercise training, as well as children and adolescents' faster recovery, reduced fatigue, lower muscular strength and intensity, and greater adaptability to aerobic activities, designing an efficient CT training program to enhance the physical fitness of this demographic has become a pressing issue (*Jansson et al., 2022*; *Mang et al., 2022*). However, the current body of research has primarily indicated that CT is linked to a reduction in several cardiovascular disease risk factors in obese children and adolescents (*Duft et al., 2020*; *Fanelli et al., 2022*; *Kelley, Kelley & Pate, 2022*). Additionally, CT offers superior benefits in terms of fat loss, body mass reduction, and cardiopulmonary function compared to isolated AT or RT (*Bouamra et al., 2022*). Nonetheless, research on the impact of CT on the physical fitness of adolescents remains limited. Only one recent study found that combining CT training with varying-intensity AT (high-intensity interval training (HIIT) *vs.* moderate-intensity continuous training (MICT)) had a similar effect on health-related physical components in adolescents (*Mendonca et al., 2022*), although the influence of intervals between AT and RT training on the development of adolescent physical fitness was not explored.

The objectives of this study were to investigate the feasibility and effectiveness of CT delivery in physical education classes at improving adolescent fitness and to compare the effectiveness of CT with different intervals of AT and RT for improving student fitness. This will provide new approaches to the design of physical education policies and programs and improving student fitness. Based on previous research (*Robineau et al., 2016*), this study hypothesized that CT delivered in a physical education class can improve students' physical fitness (muscular strength and cardiorespiratory endurance) and that CT programs that separate AT and RT may be more effective.

# MATERIALS AND METHODS

## Study design

This study employed a quasi-experimental approach with a pre-and post-test design using a control group. To ensure comparability among intervention centers, we selected three coeducational middle schools in Beijing, China, considering factors such as teacher-student ratios, teacher qualifications, experience levels, consistent educational curriculum, similar household incomes, and parental education levels. Adequate indoor and outdoor space for physical activities was also a criterion for selection.

Randomization was conducted at the class level within each intervention center. Using SPSS software, random numbers were generated, and one existing class of seventh graders was chosen at random at each center. These classes were then further randomized into three groups: CT-0h (combined AT and RT in the same class), CT-48h (AT and RT in separate sessions), and the control group (Con). Individual-level randomization within schools was avoided to prevent potential confounding effects (Weston et al., 2021).

The study involved a team of assistants responsible for data collection, sports coaches who conducted training sessions, and researchers who oversaw the training and performed statistical analyses. Data collection occurred at two time points (baseline: October 2018, outcome: January 2019), with post-intervention assessments carried out 72 h after the conclusion of the training period. Ethical approval was obtained from the Capital Institute of Sports Ethics Committee (code: 201712001, approval date: 2017/12/26), and the study was registered in the Chinese Clinical Trial Registry (ChiCTR-OOC-17014153).

## Participants

Before the study commenced, all subjects received information about the training program's risks and requirements, and informed consent was obtained from both the subjects and their guardians. A total of 163 adolescents were recruited, and the inclusion criteria for the subjects were as follows: (1) adolescents (12–13 years old), including both males and females, without any contraindications for exercise; (2) individuals who had not engaged in in-school or out-of-school sports training within the preceding 3 months; and (3) those who were not affected by any psychological disorders, such as anxiety or depression. The exclusion criteria consisted of: (1) prior participation in in-school or out-of-school sports training and (2) the presence of physical or psychological disorders.

A total of 157 individuals underwent baseline assessment (mean age: 12.48 ± 0.34 years) (Table 1). No injuries were reported during the 12-week intervention period, but seven individuals dropped out for personal reasons (two in the CT-0h group, three in the CT-48h group, and two in the control group), and their data were therefore excluded from all analyses (Fig. 1).

## Description of intervention

The intervention was applied at both the group and individual levels. Specifically, we conducted interventions for a group at a standardized time, with adjustments made to the load at the individual level based on each person's circumstances within the group. RT sessions took place in the school's designated gym facility, while AT sessions were

**Table 1  Baseline demographic characteristics of participants (*n* = 150).**

| Indicators | All (*n* = 150) | CT-0h (*n* = 50) | CT-48h (*n* = 50) | Con (*n* = 50) |
|---|---|---|---|---|
| Age (years) | 12.5 ± 0.3 | 12.6 ± 0.3 | 12.5 ± 0.4 | 12.4 ± 0.3 |
| Sex (male/female) | 67/90 | 22/30 | 23/30 | 22/30 |
| Height (cm) | 159.4 ± 7.2 | 158.6 ± 7.8 | 160.3 ± 6.6 | 159.1 ± 7.0 |
| Weight (kg) | 53.5 ± 13.0 | 52.3 ± 14.3 | 55.8 ± 11.7 | 52.3 ± 12.8 |
| BMI (kg/m$^2$) | 20.9 ± 4.1 | 20.6 ± 4.5 | 21.7 ± 3.9 | 20.5 ± 4.0 |

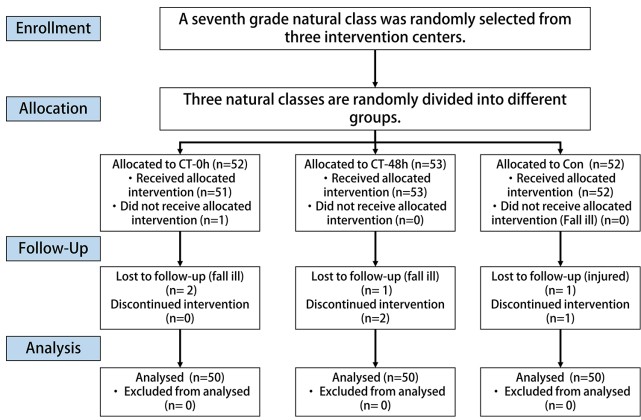

**Figure 1  Flowchart of the participant selection process.**

conducted at the school's track and field area. The research used a supervised exercise program that combined RT and AT in accordance with the WHO recommendations (*De la Corte-Rodriguez et al., 2021*). For 12 weeks, the intervention groups participated in twice-weekly physical education training sessions. A rest time of 48 h followed each training session. Each session included pre-intervention warm-up (3 min), intervention (35 min), and post-intervention stretching and relaxation (2 min) for a total of 40 min overall. All training sessions were held at the same time of day to eliminate differences due to circadian rhythms. A certified physical education teacher, along with two assistants, oversaw and led the training sessions.

### Combined training protocol

The RT intervention was meticulously crafted to align with the developmental characteristics of adolescent growth and muscle strength (*Moeskops et al., 2022*; *Myers, Beam & Fakhoury, 2017*). It was specifically designed to incorporate exercises that targeted the upper and lower limbs, as well as lumbar and abdominal movements. These exercises collectively addressed key muscle groups, including the pectoral muscles, latissimus dorsi, biceps, triceps, and quadriceps. Before the intervention, participants' one-repetition maximum (1RM) was assessed. Subsequently, the number of repetitions for each exercise was dynamically adjusted as the subject's strength progressed: 15 to 20 reps of 1RM during weeks 1 to 4, followed by 10 to 12 reps of 1RM during weeks 5 to 8, and finally, eight to 10 reps of 1RM during weeks 9 to 12 (*Stricker, Faigenbaum & McCambridge, 2020*).

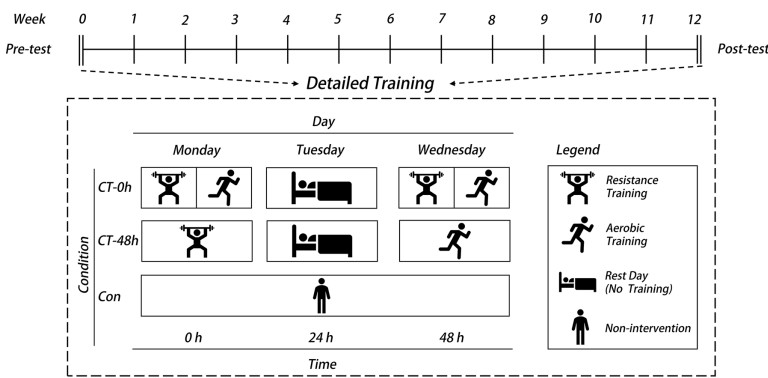

**Figure 2 Schematic diagram of the two concurrent training interventions.**

Comprehensive details of the RT protocol are available in the attached Supplemental Materials.

The AT intervention was meticulously crafted, drew on insights from a previous systematic review, and employed a HIIT approach (*Costigan et al., 2015*). This method involved running intervals of 1 min at 75–85% of HRmax, followed by 1-min intervals of rest at 50% of HRmax, with each set lasting 10 min. The participants' maximum aerobic running speed (MAS) was measured in the 4th and 8th weeks, and the running speed was adjusted based on MAS changes to maintain the AT training load at the designated heart rate. For a comprehensive understanding of the MAS testing method, please consult previous research studies (*Cao et al., 2022*). During the exercise sessions, participants' heart rates were meticulously monitored using the Mio ALPHR heart rate monitor (Petaluma, CA, United States). This device recorded heart rate changes per second during wear, ensuring that participants adhered to the prescribed exercise intensity throughout the intervention. Comprehensive details of the AT protocol are available in the attached Supplemental Materials.

### Exercise session

**CT-0h group:** This group performed the exercise programs described in the RT and AT protocols in the same session, with one set of RT followed by one set of AT. The training was performed twice a week for 35 to 40 min each session.

**CT-48h group:** This group performed the exercise programs described in the RT and AT protocols in two physical education classes per week, respectively. Two sets of RT were performed on Mondays and two sets of AT were performed on Wednesdays, with 35–40 min of training each time.

**Con group:** This group maintained their normal daily activities without intervention (Fig. 2).

### Study measurements

#### Procedure

We measured the height and weight of each student before and after the intervention using a body composition analyzer (InBody J20; BIOSPACE, Seoul, Korea), known for its strong

validity in Asian children and utilization in large-scale intervention trials with children (*Kriemler et al., 2010*). Additionally, we calculated the body mass index (BMI) as weight in kilograms divided by the square of the height in meters.

We executed the measurement of pertinent parameters during two distinct time frames: 1 week preceding the intervention and the initial week following the intervention. Our deliberate choice to distribute these evaluations across multiple days aimed to alleviate the potential influence of participant fatigue and maintain the precision of measurements. The procedural chronology for these assessments was delineated as follows:

**Body composition analysis:** This assessment was administered on the inaugural day, prior to the commencement of any physical activities.

**Primary outcome assessments:** These included evaluations of CRF through the estimated $VO_2$max test and assessments of muscle strength, including the standing long jump, vertical jump, and handgrip strength test (*Cadenas-Sanchez et al., 2016*; *Janz et al., 2021*). We meticulously ensured that each of these assessments was conducted on distinct days in order to uphold the accuracy of results while minimizing the risk of participant fatigue.

**Secondary outcome evaluations:** These included the 30-s sit-up test, 50 m sprint, and aerobic capacity index, with distances of 1,000 m for boys and 800 m for girls. In consonance with the primary outcome tests, these assessments were conducted on separate days, serving to comprehensively characterize the participants' physical fitness profiles.

### Primary outcome

In this study, we considered CRF and muscle strength as the primary outcome indicators, as they are direct indicators of health in adolescents (*Loo et al., 2022*).

## Estimated $VO_2$max

We used the multistage 20 m shuttle run test, which is used to estimate $VO_2$max (*Leger et al., 1988*). This test involved running back and forth on a 20 m track with an audio signal until exhaustion. $VO_2$max was estimated using a specific formula suitable for adolescents aged 10–16 years (*Cadenas-Sanchez et al., 2016*; *Lang, 2018*).

## Muscle strength tests

Standing long jump: Participants performed three jumps, and the best jump length (measured in centimeters) was recorded.

Handgrip strength: The data collection process was carried out according to the methodology described in *Li et al. (2023)*. The handgrip strength of the subjects was measured using a pneumatic squeeze bulb dynamometer (Baseline, White Plains, NY, USA) while seated and with their elbows resting on a table. Two unilateral grip strength measurements were taken consecutively, with the first measurement being taken on the dominant hand. A 30-s rest period was introduced between each measurement to prevent fatigue. From the two trials, the best value for each hand was selected.

*Secondary outcomes*

We assessed various secondary outcomes to gain a comprehensive understanding of the students' physical fitness.

Muscular endurance (30-s sit-up test): For the 30-s sit-up test, participants were instructed to lie on their backs with their knees bent and feet flat on the ground. Their hands were placed behind their heads with fingers interlocked to support the neck. Participants then performed sit-ups by raising their upper bodies until their elbows touched their knees and then returned to the starting position. The number of correctly completed sit-ups within the 30-s time frame was recorded (*Janz et al., 2021*).

Speed and anaerobic capacity (50 m sprint): Participants sprinted as quickly as possible along a 50 m track, and the running time in seconds was recorded. The experimenter timed the tests with a stopwatch used in track races (Casio, Tokyo, Japan), which was accurate to 0.01 s (*Fjortoft et al., 2011*).

Aerobic capacity index: For boys and girls, we measured the aerobic capacity index using a 1,000 m and 800 m run, respectively, following the Chinese National Students' Physical Fitness and Health Standard.

Each test was carefully selected based on its relevance and validity in assessing specific aspects of the students' physical fitness.

## Statistical analysis

We estimated the sample size using G*Power 3.1. Based on a previous study (*Alves et al., 2016b*), we set the test efficacy (1-$\beta$) to 0.80, incidence of type I error $\alpha$ to 0.05, correlation between pre-and post-intervention to 0.80, and effect size to 2.68. The sample size required for each group was calculated to be 38. To account for a potential drop-out rate of 20%, we have determined that each group should consist of at least 48 subjects, so the total number of subjects should not be less than 144, and 157 subjects were recruited.

The collected data were analyzed using IBM SPSS Statistics 26 software. Descriptive statistics, including the mean ± SD, were used to summarize the data. The normal distribution of variables was assessed using the Shapiro-Wilk test, while the homogeneity of variances was examined using the Levene test. To evaluate the differences associated with demographic variables (such as age, height, weight, and BMI), a one-way analysis of variance was conducted. The chi-square test was employed to assess gender differences.

For the experimental groups, a 3 (group: CT-0h, CT-48h, and control) × 2 (time: pre- and post-test) repeated measures analysis of variance was utilized to examine the effect of the intervention before and after implementation. Confounding variables (baseline, sex, BMI, and age) were controlled for by analysis of covariance, and outcome indicators were compared across groups (Bonferroni). In addition, the effect size with statistical significance was calculated, which was expressed as $\eta_p^2$ and Cohen's d, in which a small effect was 0.01/0.20, a medium effect was 0.06/0.50, and a large effect was 0.14/0.80 (*Batista et al., 2013*; *Richardson, 2011*). The significance level was set to $p < 0.05$.

## RESULTS

### Effects of intervention on primary fitness outcomes

The outcomes from the intra-group analysis are detailed in Table 2. Utilizing a repeated-measures ANOVA, notable enhancements were observed in key fitness parameters, signifying the intervention's efficacy. These parameters include Estimated VO2max ($F_{(2,150)}$ = 6.505, $p$ = 0.002, $\eta_p^2$ = 0.063), 20 m shuttle run (laps) ($F_{(2,150)}$ = 20.216, $p < 0.001$, $\eta_p^2$ = 0.216), standing long jump ($F_{(2,150)}$ = 4.098, $p$ = 0.019, $\eta_p^2$ = 0.053), and handgrip strength ($F_{(2,150)}$ = 8.23, $p < 0.001$, $\eta_p^2$ = 0.042). These results indicate significant main effects of the interaction between group and time. More specifically, subsequent to the intervention, notable improvements were recorded in the 20 m shuttle run (laps), standing long jump, and handgrip strength across both the CT-0h and CT-48h groups ($p < 0.05$). The CT-0h group alone demonstrated significant enhancements in Estimated VO2max ($p < 0.05$).

Further analysis, adjusting for baseline variables such as sex, Baseline data, and age through covariance analysis, highlighted significant differences between groups. These findings, depicted in Fig. 3, revealed pronounced disparities in Estimated VO2max ($F_{(2,150)}$ = 38.038, $p < 0.001$, $\eta_p^2$ = 0.569), 20 m shuttle run ($F_{(2,150)}$ = 124.52, $p < 0.001$, $\eta_p^2$ = 0.812), standing long jump ($F_{(2,150)}$ = 104.598, $p < 0.001$, $\eta_p^2$ = 0.784), and handgrip strength ($F_{(2,150)}$ = 74.25, $p < 0.001$, $\eta_p^2$ = 0.721).

In particular, the CT-0h group exhibited significantly greater improvements compared to the control (Con) group ($p < 0.05$) in all assessed parameters (Fig. 3): Estimated VO2max, 20 m shuttle run (laps), standing long jump, vertical jump, and handgrip strength. The CT-48h group also showed superior enhancements relative to the Con group ($p < 0.05$) in the same parameters, excluding Estimated VO2max. Differences between the CT-0h and CT-48h groups were statistically significant in Estimated VO2max and 20 m shuttle run (laps) ($p < 0.05$).

### Intervention impact on secondary fitness outcomes

The analysis of intra-group comparisons, as outlined in Table 2, demonstrates significant improvements in secondary fitness outcomes including sit-ups ($F_{(2,150)}$ = 12.41, $p < 0.001$, $\eta_p^2$ = 0.141), 50 m sprint ($F_{(2,150)}$ = 5.35, $p$ = 0.006, $\eta_p^2$ = 0.068), 800 m run ($F_{(2,150)}$ = 16.76, $p < 0.001$, $\eta_p^2$ = 0.278), and 1,000 m run ($F_{(2,150)}$ = 10.266, $p < 0.001$, $\eta_p^2$ = 0.265). These results suggest significant interaction effects between group and time. After the intervention, both the CT-0h and CT-48h groups demonstrated significant improvement in sit-ups, 800-m run, and 1,000-m run ($p < 0.01$). Additionally, the CT-0h group exhibited a noteworthy enhancement in the 50-m sprint ($p < 0.01$), whereas the Con group did not display any significant changes in the various tests ($p > 0.05$).

Covariance analysis, adjusted for confounders, revealed that post-intervention outcomes of both CT-0h and CT-48h groups were significantly superior to those of the Con group ($p < 0.05$), with the exception of the 50-m sprint. A noteworthy distinction was observed between the CT-0h group and the CT-48h group in the 50-m sprint (Mean Difference: 0.30 s; 95% CI [−0.60 to −0.002]; $P$ = 0.048), while no significant variances were

**Table 2 Intra-group comparison of the effects of 12 weeks of training on participants' physical fitness (mean ± sd.).**

| Variables | Group | Pre | Post | Change (95%CI) | p-value | Cohen's d | ANOVA p-value ($\eta_p^2$) | | |
|---|---|---|---|---|---|---|---|---|---|
| | | | | | | | Time | Group | T × G |
| VO2max (mL/kg/min) | CT-0h | 37.5 ± 4.8 | 39.3 ± 4.9 | 1.8 [1.0–2.6] | <0.001 | 0.37 | 0.002 (0.06) | 0.196 (0.02) | 0.009 (0.06) |
| | CT-48h | 36.7 ± 3.4 | 37.3 ± 3.6 | 0.6 [−0.3 to 1.4] | 0.195 | — | | | |
| | Con | 37.5 ± 4.2 | 37.5 ± 3.4 | −0.05 [−0.9 to 0.8] | 0.898 | — | | | |
| PACER (laps) | CT-0h | 33.5 ± 16.9 | 45.3 ± 18.3 | 11.7 [9.8–13.8] | <0.001 | 0.67 | <0.001 (0.52) | 0.078 (0.034) | <0.001 (0.22) |
| | CT-48h | 30.0 ± 13.0 | 37.7 ± 14.0 | 7.8 [5.8–9.8] | <0.001 | 0.57 | | | |
| | Con | 32.1 ± 14.9 | 34.7 ± 13.0 | 2.6 [−0.6 to 4.7] | 0.105 | — | | | |
| Standing long jump (cm) | CT-0h | 165.6 ± 20.2 | 172.5 ± 20.7 | 6.9 [4.1–9.7] | <0.001 | 0.34 | <0.001 (0.12) | 0.015 (0.056) | 0.019 (0.053) |
| | CT-48h | 159.1 ± 20.7 | 165.4 ± 21.4 | 6.2 [0.4–12.1] | <0.001 | 0.26 | | | |
| | Con | 157.6 ± 17.6 | 159.2 ± 18.7 | 1.7 [−1.2 to 4.5] | 0.247 | — | | | |
| Handgrip strength (kg) | CT-0h | 23.2 ± 5.4 | 33.9 ± 6.7 | 10.7 [9.3–12.1] | <0.001 | 1.8 | <0.001 (0.57) | <0.001 (0.26) | <0.001 (0.44) |
| | CT-48h | 23.3 ± 4.5 | 31.8 ± 6.8 | 8.5 [6.8–10.1] | <0.001 | 1.1 | | | |
| | Con | 21.7 ± 4.7 | 21.6 ± 4.2 | −0.1 [−1.5 to 1.3] | 0.175 | — | | | |
| Sit-ups (reps) | CT-0h | 37.7 ± 9.7 | 45.0 ± 8.3 | 7.3 [5.6–9.0] | <0.001 | 0.81 | <0.001 (0.37) | 0.038 (0.044) | <0.001 (0.14) |
| | CT-48h | 37.2 ± 8.0 | 42.1 ± 7.5 | 4.9 [3.2–6.6] | <0.001 | 0.63 | | | |
| | Con | 36.7 ± 8.2 | 38.0 ± 7.7 | 1.4 [−0.3 to 3.0] | 0.111 | — | | | |
| 50-m sprint (s) | CT-0h | 9.2 ± 0.6 | 8.9 ± 0.7 | −0.3 [−0.5 to −0.2] | <0.001 | −0.52 | 0.057 (0.025) | 0.007 (0.066) | 0.006 (0.068) |
| | CT-48h | 9.3 ± 1.0 | 9.3 ± 1.3 | −0.02 [−0.2 to 0.2] | 0.792 | — | | | |
| | Con | 9.6 ± 0.8 | 9.6 ± 0.6 | 0.06 [−0.1 to 0.2] | 0.509 | — | | | |
| 800 m (s) | CT-0h | 263.3 ± 35.2 | 225.9 ± 21.2 | −37.4 [−46.9 to −27.8] | <0.001 | −1.29 | <0.001 (0.51) | 0.025 (0.082) | <0.001 (0.28) |
| | CT-48h | 281.4 ± 43.0 | 244.1 ± 24.9 | −37.3 [−46.8 to −27.8] | <0.001 | −1.06 | | | |
| | Con | 266.2 ± 33.5 | 262.7 ± 37.3 | −3.4 [−12.9 to 6.1] | 0.475 | — | | | |
| 1000 m (s) | CT-0h | 326.1 ± 60.8 | 292.9 ± 45.9 | −33.3 [−46.3 to −20.2] | <0.001 | −0.62 | <0.001 (0.28) | 0.078 (0.086) | <0.001 (0.27) |
| | CT-48h | 322.7 ± 49.3 | 297.0 ± 39.6 | −25.7 [−38.7 to −12.6] | <0.001 | −0.57 | | | |
| | Con | 336.2 ± 52.1 | 342.4 ± 44.1 | 6.2 [−6.9 to 19.3] | 0.347 | — | | | |

**Note:**
CT-0h refers to the combination of resistance training (RT) and aerobic training (AT) in the same physical education (PE) session, and CT-48h refers to RT and AT separated between two PE sessions (48 h interval). The p-value represents the outcome of pairwise comparisons conducted pre- and post-intervention. The ANOVA p-value pertains to the repeated measures of variance, while T × G signifies the interaction effect between time and group.

noted in other performance metrics ($P > 0.05$). Moreover, there were no significant variations identified between the CT-48h group and the Con group in the 50-m sprint ($P > 0.05$) (Fig. 3).

# DISCUSSION

Adolescents taking part in physical education classes frequently perform RT and AT concurrently due to various limitations in the school setting (*e.g.*, little practice time or lack of facilities) (*Santos et al., 2012*). This study compared how two CT protocols affected adolescents' physical fitness, with RT and AT scheduled for either the same session (CT-0h group) or separate sessions (CT-48h group). The study's findings indicated that both CT programs were successful in increasing adolescent fitness, but that the program combining RT and AT in one session appeared to be more effective at improving adolescent speed indices and estimated $VO_2$max. The hypothesis of this study was rejected.

First, the CT-48h group showed no improvement in estimated $VO_2$max despite an increase in 20 m shuttle run (laps), which earlier research had suggested was moderately

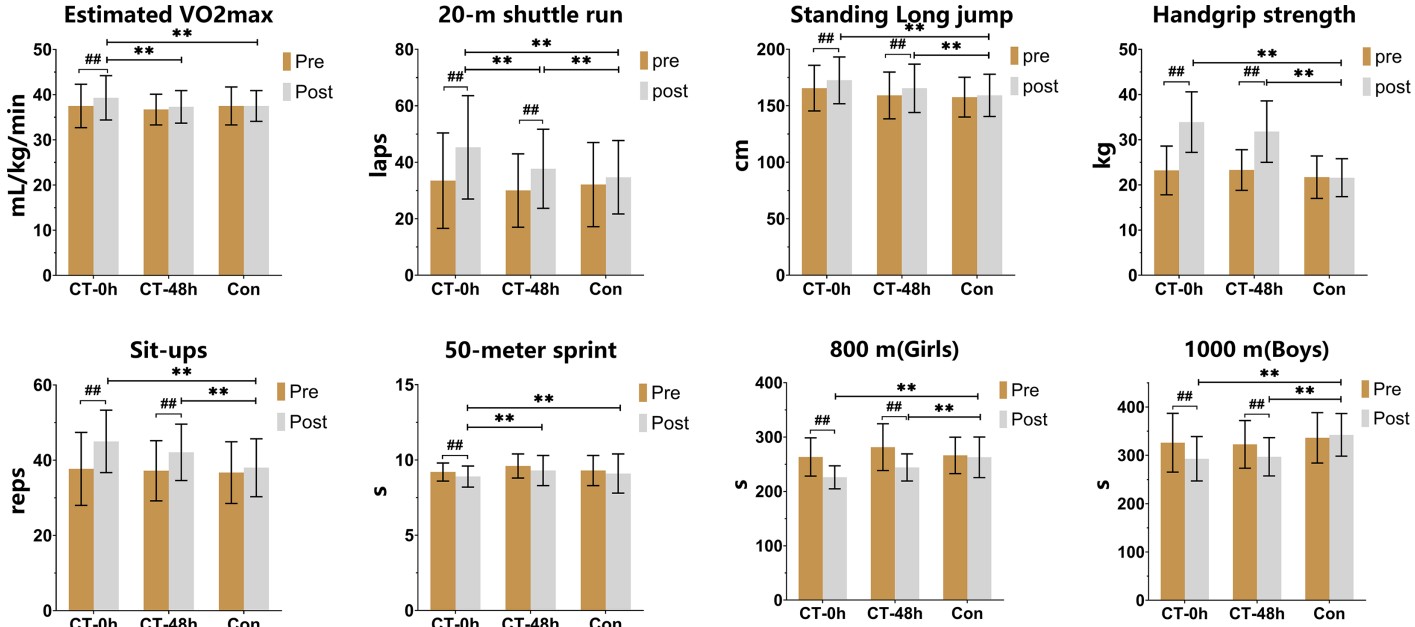

**Figure 3** **Comparison of the effects of 12 weeks of training on participants' physical fitness (mean ± SD).** All comparisons were adjusted for the subjects' sex, baseline data, and age for each indicator. ##Significant within-group changes from pre- to post-training (repeated measures ANOVA results), $p < 0.001$; **Significantly different between groups (covariance results), $p < 0.001$.     

correlated with maximal oxygen consumption (VO$_2$peak) ($R^2 = 0.49$) (*Lang, 2018*). This finding may be due to the fact that the CT-48h intervention protocol was less successful at enhancing cardiorespiratory endurance. The between-group comparison concluded that the protocol that included RT and AT in the same session was superior to the protocol presenting the exercises separately at improving estimated VO$_2$max in adolescents. This is inconsistent with the findings of *Robineau et al. (2016)*, who concluded that performing AT and RT in the same training session (no interval) or in 1 day (interval <6 h) was not ideal for aerobic capacity improvement, but better adaptation changes were observed when training was performed on alternate days (interval >24 h). The reason for the difference may be related to the intensity of AT, which was more intense (120% maximal aerobic velocity) in the HIIT protocol used in the study by *Robineau et al. (2016)* than in this study. A different study showed that when implementing a CT protocol, selecting a lower AT intensity induced a higher aerobic adaptation capacity (*Sousa et al., 2019*). In addition, the choice of subjects can also produce differences in study results; *Robineau et al.'s (2016)* subjects were athletes, while the present study included general adolescents. A meta-analysis showed that CT produces different adaptations in athletes and non-athletes (*Petre et al., 2021*). *Santos et al. (2012)* also found, in a population of children and adolescents, that estimated VO$_2$max improved only when RT and AT were performed simultaneously within the class. This outcome can be explained by the fact that in some ways, combining AT and RT in one session can enhance angiogenesis and oxidase activity (such as capillary to fiber area and succinate dehydrogenase (SDH) activity) more than strength or endurance training alone (*Grgic et al., 2019*), and is, therefore, better for CRF. Considering the importance of CRF levels for the present and future health of adolescents,

combining RT and AT into the same sessions should be considered in order to optimize the physical education class schedule in schools.

In addition, previous studies have suggested that CT significantly reduces the development of strength and muscle hypertrophy compared to RT alone, which is often referred to as the "interference effect" (*Camera, 2022*; *Hickson, 1980*). Recently, a growing body of evidence has demonstrated that CT does not impair strength development in prepubertal adolescents (*Alves et al., 2018*, *2016a*; *Bouamra et al., 2022*; *Gabler et al., 2018*; *Jansson et al., 2022*), possibly because adolescents have different neuromuscular and endurance characteristics than adults (*Marta et al., 2013*), such as their lack of muscle hypertrophy in response to RT (*Gabler et al., 2018*). However, current research is still limited in that it ignores the variable of the training interval between RT and AT. We concluded that the length of the training interval did not have a significant effect on the development of strength in adolescents and that there was no difference between the two CT protocols in improving muscle strength in adolescents. The findings of this study disprove the previous theory that a training interval of less than 6 h between RT and AT affects muscle strength and hypertrophy (*Petre et al., 2021*; *Robineau et al., 2016*), at least among teenagers.

The AT training method chosen for this study and the training order between AT and RT may be the cause for the training interval between AT and RT not affecting muscle strength in adolescents. It was previously concluded that medium-intensity and long-duration endurance training inhibits the activation of the protein kinase B–mammalian rapamycin pathway and adenosine monophosphate-activated protein kinase (AMPK), which in turn negatively affects the development of muscle strength (*Methenitis, 2018*). While the present study used the HIIT form of AT, previous studies have found that HIIT shares the same metabolic adaptations as RT, such as activating the receptor-γ coactivator (PGC-1a) pathway *via* the peroxisome proliferator, which contributes to an increase in muscle strength. Recently, it was also shown that HIIT does not have a disruptive effect on muscle strength improvement (*Vechin et al., 2021*). Therefore, it is reasonable to assume that performing AT in the same session using CT based on HIIT, an AT protocol, will not impair any increase in muscle strength. The "chronic interference hypothesis", another explanation for the "interference effect", which contends that performing aerobic endurance training first in a session can negatively impact the effectiveness of RT and cause excessive fatigue that is harmful to muscle recovery (*Lee et al., 2020*). The sequencing used in this study, however, put RT before AT, which has been proven to be more favorable for lower body strength adaptation (*Murlasits, Kneffel & Thalib, 2018*).

In addition to muscle strength and explosive power, we observed significant changes in abdominal muscle endurance (sit-ups) in both training groups after 12 weeks of intervention, a positive result that is significant because muscle fitness (strength, explosive power, and endurance) is negatively associated with clustered cardiometabolic risk, obesity (*Silva et al., 2020*), and positively associated with skeletal health and self-esteem (including adolescents' physical self-concept, perceived physical appearance, and perceived athletic

ability) (*Smith et al., 2014*). Therefore, actively encouraging adolescents to participate in RT or CT is essential to maintain and/or improve muscle fitness.

While sports performance, which is dependent on muscle strength, is as crucial to preserving health as aerobic capacity, many recent studies on physical fitness have concentrated on aerobic capacity (*Alves et al., 2016a*; *Taanila et al., 2011*). This study concluded that CT was very effective at improving adolescent sports performance (1,000 m and 800 m run). Previous studies have concluded that CT is more effective than unimodal AT in terms of sports performance. This is due to the fact that RT increases musculotendinous unit stiffness, which leads to an increased ability of the muscle to store elastic energy in tandem and parallel elastic components when doing centrifugal movements. This in turn increases the strength of the muscle when doing centripetal movements, which can improve sports economy (*Bazyler et al., 2015*; *Gabler et al., 2018*). The improvement of cardiorespiratory endurance and exercise economy is the basis for improving sports performance and endurance (*Gabler et al., 2018*). Additionally, this study discovered that CT-0h improved the speed index more effectively than CT-48h. Speed is positively correlated with maximal oxygen uptake and leg strength (*Suchomel, Nimphius & Stone, 2016*).

## Study strengths and limitations

The results of this study demonstrate the possibility for integrating CT into school physical education lessons to enhance adolescent health. This study could act as a guide for scheduling CT of strength and aerobic exercise components in physical education classes in schools and offers scientific support for the development of instructional content in middle school physical education programs. To our knowledge, this study is the first to consider how CT training protocols with different training intervals between RT and AT affect physical fitness in adolescents, and the results can be used to enrich and develop theories related to CT training in adolescents. In practice, twice-weekly CT of either protocol has been shown to improve muscle fitness in adolescents; however, the CT-0h program without training intervals appeared to be more effective at improving CRF. This serves as a useful resource for developing exercise plans and establishing objectives for health promotion in the adolescent population.

However, there are some limitations to this study. First, due to its implementation in a school physical education class, it was not possible to perform a complete random assignment of each student. However, minor differences in performance abilities between the three groups were considered acceptable, as observed in previous studies conducted in a school setting (*Bossmann, Woll & Wagner, 2022*). Second, this study did not assess the affective (*i.e.*, sensory state) and enjoyment responses of adolescents participating in the two CT protocols. This aspect should be addressed in future studies, as perceived responses are known to correlate with habitual physical activity and exercise adherence throughout an individual's lifespan (*Ricci et al., 2020*). Additionally, this study overlooked the measurement of participants' "maturation," which may introduce a certain degree of variability. Future research should further investigate the impact of adolescents' "maturation" on the effectiveness of interventions. Finally, it is worth noting that only a

running format was employed for AT in this study. Whether other AT protocols (*e.g.*, cycling) have different effects on the outcomes of CT intervention remains unknown and warrants exploration in future research.

## CONCLUSION

This study confirmed that both studied CT programs delivered in school physical education classes were effective at improving physical fitness indicators in adolescents. A program that schedules RT and AT in the same class, with RT followed by aerobic endurance training, may result in greater gains in CRF. Future studies could expand the study size and consider psychological factors and variables such as type, intensity, and duration of AT training, as well as control for habitual behaviors related to physical activity and food consumption during the intervention.

### Funding

This work was funded by the National Key Research and Development Program of China (Grant No. 2020YFC2006200). The funders had no role in study design, data collection and analysis, decision to publish, or preparation of the manuscript.

### Grant Disclosures

The following grant information was disclosed by the authors:
National Key Research and Development Program: Grant No. 2020YFC2006200.

### Competing Interests

The authors declare that they have no competing interests.

### Author Contributions

- Zhen Li conceived and designed the experiments, performed the experiments, analyzed the data, prepared figures and/or tables, authored or reviewed drafts of the article, and approved the final draft.
- Teng Ding conceived and designed the experiments, performed the experiments, authored or reviewed drafts of the article, and approved the final draft.
- Yanan Gao conceived and designed the experiments, authored or reviewed drafts of the article, and approved the final draft.
- Xiaowei Han analyzed the data, authored or reviewed drafts of the article, and approved the final draft.
- Yang Liu performed the experiments, analyzed the data, authored or reviewed drafts of the article, and approved the final draft.
- Zhixiong Zhou conceived and designed the experiments, authored or reviewed drafts of the article, and approved the final draft.

## Human Ethics

The following information was supplied relating to ethical approvals (*i.e.*, approving body and any reference numbers):

The Ethics Committee of the Capital University of Physical Education And Sports approved the study (code: 201712001, approval date: 2017/12/26).

## Data Availability

The raw data are available in the Supplemental File.

## Clinical Trial Registration

The following information was supplied regarding Clinical Trial registration:

ChiCTR-OOC-17014153.

## Supplemental Information

Supplemental information for this article can be found online at http://dx.doi.org/10.7717/peerj.17294#supplemental-information.

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
