# Peer review of "A comparison of the effects of two protocols of concurrent resistance and aerobic training on physical fitness in middle school students"

_PeerJ, doi:10.7717/peerj.17294_

## Round 0.1 · original submission · Major Revisions

Dear authors, the reviewers pointed out several concerns and suggestions that must be addressed before further evaluation. Please, consider their comments, especially regarding the training protocols description, statistic issues, and the results presentation. The use of specific tables and figures is encouraged, respectively.

**Language Note:** PeerJ staff have identified that the English language needs to be improved. When you prepare your next revision, please either (i) have a colleague who is proficient in English and familiar with the subject matter review your manuscript, or (ii) contact a professional editing service to review your manuscript. PeerJ can provide language editing services - you can contact us at copyediting@peerj.com for pricing (be sure to provide your manuscript number and title). – PeerJ Staff

·

Basic reporting

Basic reporteining
It is very good to highlight the importance of muscular strength in health in the introduction, and this reviewer agree that CT is an interesting training method to study. The "interference effect" is exciting and typically examined in adults, which the text also states. However, I would encourage authors to be more specific about why studying in children/adolescents is important. For instance, Gäbler et al. 2018 (Front Physiol) hypothesized that the interference effect is age dependent and that children probably have a potentiation effect of CT rather than an interference effect. Child vs. adult physiology differs in many ways e.g., children recover quicker, fatigue less, lower muscle power and strength and also is more adapted to aerobic activities etc., which may be good info to add to why studying CT in these age groups is interesting. I also think the introduction could be improved with more previous research on CT effects in children and adolescents (see Gäbler et al. 2018 for review). CT vs. single-mode training has been studied and is relevant to add what this specific paper adds are the comparisons of within vs between session CT which is a good knowledge gap and specifically since it also includes females.

– Abstract: Only use abbreviation when it is necessary. Removing abbreviations makes it easier for the reader to follow. It should also be spelled out in full the first time used. Abstract also lack subheadings


2.1 Lines 109-130: It is hard to follow the text flow in the study design. I think it can be shortened a lot and try to be more concise. It is a 12-week training study with three groups with tests before and after the training period. The word "natural classes" might not be the correct terminology.

Line 124-126: This part is difficult to understand. How is it possible to use blinding in a training study? If coaches supervise a program, they know which group does which training. Please clarify or consider remove.

Line 142-144: inclusion criteria 1 should include "male and female". Inclusion criteria 3: please clarify or give examples for disorders.

Line 152: difficult to understand. Please clarify the meaning of it.

Table 1. Please remove non-relevant decimals from all tables. One decimal is good enough. Same in Table 2

Table 2 All abbreviations should be put as footnotes with names spelled out. VO2 max and PACER should be replaced with 20-m shuttle run (estimated VO2max) and 20-m shuttle run (laps)

Experimental design

Experimental design
This study examines the effects of 12 weeks of within-session concurrent training versus between-session concurrent training on measures of aerobic fitness, muscle strength, power, and sprint ability in male and female adolescents. The study is well designed and executed, with a substantial sample size that encompasses both genders within the adolescent demographic. The findings distinctly reveal how concurrent training influences various aspects of athletic performance when contrasted with a control group. One of the study's notable strengths lies in its incorporation of a relatively intricate training design (concurrent training) that yields enduring fitness improvements within a pivotal age group, set within a school-based context.

Method: The description of the intervention is probably appropriate but should be rewritten with a focus on more details so researchers can interpret the program quality and replicate it. It would be beneficial with a table with the training program, including all relevant details such as exercises used, reps, set, intensity, and rest periods. It is also unclear how the loads in the resistance program were selected. Were there any 1RM tests? Similarly, the aerobic training program is not explained in enough detail. The authors used both MAS and HRmax, but it is unclear which one was used to determine exercise intensity. Also, which exercise test was used to determine MAS and HRmax? Please clarify.
Methods: Primary outcome and secondary outcome include the procedure of how the test was conducted. I suggest separating them into something similar to "procedure," which in detail explains how the test for the test was conducted. Primary and secondary outcome text can shortly explain which parameters were included. Please also add info on the validity and reliability of each test used. The procedure should also include information on how many test days were used and in which orders the tests were conducted.

Line 132-135: Suggest moving these to the "statistics" section

Line 203: In this paper, the authors estimate VO2max through a 20-m shuttle run test and don't use direct measurements of VO2 max. Please be consistent in text and use the "estimated VO2max" instead of just "VO2max," which could be misleading. Also, suggest not using the term CRF since 20-m shuttle run validity has been questioned as a marker of CRF (please see Welsman & Armstrong 2019). Lastly, be consistent in the terminology, use, and the term "20-m shuttle run" instead of PACER. I think that is the most commonly adopted terminology.

Line 221: This is probably a unilateral test, not a bilateral one. Please clarify.

Line 229: How was this test measured? Photocells or maybe just a stopwatch? Please add information.

Validity of the findings

Statistics: overall stats need to be double-checked. See other comments

A primary concern arises with the authors' conclusion, as articulated in line 40, asserting that "the program that included resistance training and aerobic training in the same physical education class was better at improving the CRF of adolescents." Upon scrutiny of Table 2, it is evident that the CT-0h group displayed an increase from 37.5 (SD 4.76) to 39.3 (SD 4.90), whereas CT-48h exhibited a rise from 36.7 (SD 3.7) to 37.3 (SD 3.6). The substantial variability raises doubts about the statistical significance of the observed difference. It is recommended that a standard bar plot be employed to visually depict this minute disparity, potentially elucidating the matter. A re-evaluation of the statistical analysis is advisable. Additionally, clarification regarding whether the values in Table 3 represent means and standard deviations is warranted.

Results: It would be useful to highlight some of the most important results in a plot instead of putting all data in tables. It is a quick way for the reader to visualize the effects of training. Possibly the main outcomes are most interesting.

Line 239-243: The authors seem to use ANCOVA in addition to ANOVA. However, no results seem to be reported. ANCOVA produces adjusted means that aren't reported anywhere in the result section.

The result section in general, is difficult to read. The most important is to clearly report if there are any statistically significant differences between the experimental groups and experimental groups vs control groups. Adding effect size is also a good idea. Reference to table 2 should be in first sentence. Authors seem to mix up between-group differences and within-group differences.

282-284: Please double-check statistics. I doubt that there is a statistically significant difference between the experimental groups in sprint ability and estimated VO2 max. If the authors plot data in a standard bar graph, it should be evident that the SD is high.
Line 366: one important aspect missing in this study design is a measure of “maturation” such as Tanner or peak-heigh velocity (PHV). This should be addressed and discussed mainly since this study includes participants that could be in different maturation stages, which might increase variability and, thereby, the results and interpretation of the study.

Reviewer 2 ·

Basic reporting

No comment

Experimental design

My only question here would be the choice to measure muscular strength with traditionally power-oriented tests such as vertical and broad jump. Hand grip dynamometry makes sense, but I would prefer to see the other muscular strength measures more closely aligned with the resistance training protocol itself, such as 1RM or 3- to 5RM tests of strength in a movement(s) performed by the participants during their training (chest press, leg press, etc.).

With this in mind, I'd like to see the authors defend their choice to use muscular power tests in more detail

Validity of the findings

No comment

Additional comments

This paper addresses an interesting and important topic with relevance for adolescent health, fitness, and possible planning of physical education class time. Methodology was conducted in a rigorous fashion, and results are explained clearly and with appropriate support from current research. I applaud the authors on a very well conducted study!

While I don't necessarily think it has to be included in the discussion, a question came to mind about possible within-session motivation for the participants. Given the age of the students, do the authors think the novelty of the CT-0h protocol could have played a role in the students providing more effort during those training sessions as compared to the CT-48h group? I know effort/motivation was not measured, so it would be conjecture, but perhaps something to be explored in future work.

---

## Round 0.2 · Minor Revisions

Dear authors, thank you for addressing the reviewer's suggestions. Please, consider the remaining issues raised and resubmit the manuscript.

·

Basic reporting

ok

Experimental design

I thank the authors on the revised paper. I think most of my comments have been revised. I have a few suggestions on how to improve the paper further.
1. Method section: I think the method section has been improved. The description of the selected tests are, in my opinion, still very broadly described and could be improved even more with more details. How each test was standardized can be written with more clarity. For instance,” vertical jump” – how was this test standardized, and which equipment was used to assess flight height? Did the authors use a CMJ or SJ test? (with or without a countermovement jump). Sit-ups are another test that can be described in more detail; one example is where the hands were placed during the test. Did the authors standardize the velocity?

Validity of the findings

2. Table 2 lacks a description of what ”a” and ”b” refer to. I assume this table is based on absolute values (e.g no adjusted values).
3. Figure 3: I thank the authors for using figures to visualize data. Typically, a figure for a training study has pre and post-data for each group and test so the reader can quickly visualize how much ”impact” the training had for each group. This figure seems only to contain” baseline” values, which don't help the reader much. I encourage the author to change it. Also, according to the result section, this figure should be based on adjusted means (from an ANCOVA), but the figure does not mention it. Is the data adjusted mean or just raw data?
Lastly, if the authors decide to make a pre-post figure, maybe only highlight those data that are of particular interest to the reader (e.g., it is not necessary to make a figure with all outcomes).

---

## Author Rebuttal · Round 0.2

# A Comparison of the Effects of Two Protocols of Concurrent Resistance and Aerobic Training on Physical Fitness in Middle School Students

Zhen Li, [1†]Teng Ding, [2†] Yanan Gao,[1] Xiaowei Han,[3] Yang Liu,[4] and Zhixiong Zhou [5]

[1]School of Physical Education and Sport Sciences, Fujian Normal University, Fuzhou, 350117, P.R. China

[2]School of Physical Education, Ningxia University, Yinchuan 750021, P. R. China

[3]Faculty of Education, Beijing Normal University, Beijing 100875, P. R. China

[4]Hebei institute of international business and economics, Qinhuangdao 066100, P. R. China

[5]School of Physical Education and Coaching Science, Capital University of Physical Education and Sports, Beijing 100191, P. R. China

†Equal contributors

Corresponding Author:

Zhixiong Zhou[5]

11 North Third Ring Road West, Haidian District, Beijing, 100191, P. R. China

Email address: zhouzhixiong@cupes.edu.cn

## Your peer-review comments:

## Editor's Decision MAJOR REVISIONS

Dear authors, the reviewers pointed out several concerns and suggestions that must be addressed before further evaluation. Please, consider their comments, especially regarding the training protocols description, statistic issues, and the results presentation. The use of specific tables and figures is encouraged, respectively.

**PeerJ Staff Note:** Please ensure that all review, editorial, and staff comments are addressed in a response letter and any edits or clarifications mentioned in the letter are also inserted into the revised manuscript where appropriate.

**Language Note:** PeerJ staff have identified that the English language needs to be improved. When you prepare your next revision, please either (i) have a colleague who is proficient in English and familiar with the subject matter review your manuscript, or (ii) contact a professional editing service to review your manuscript. PeerJ can provide language editing services - you can contact us at copyediting@peerj.com for pricing (be sure to provide your manuscript number and title). – PeerJ Staff

*Dear PeerJ Staff and Reviewers,*

*We appreciate the thorough review and constructive feedback provided on our manuscript. We are committed to addressing all concerns raised and improving the overall quality of the manuscript.*

*We take note of the language concerns highlighted by PeerJ staff. In our revision, we will either engage a proficient colleague with a good command of English and subject knowledge or consider professional editing services to enhance the manuscript's language quality.*

*We assure the PeerJ staff that all comments, including those from reviewers and editorial notes, will be diligently addressed in our revised manuscript. We are committed to delivering a refined version that aligns with the journal's standards.*

*Sincerely,*
*Zhen Li*

# Comments from the reviewers

## Reviewer: Daniel Jansson

**Basic reporting**

Basic reporteining

It is very good to highlight the importance of muscular strength in health in the introduction, and this reviewer agree that CT is an interesting training method to study. The "interference effect" is exciting and typically examined in adults, which the text also states. However, I would encourage authors to be more specific about why studying in children/adolescents is important. For instance, Gäbler et al. 2018 (Front Physiol) hypothesized that the interference effect is age dependent and that children probably have a potentiation effect of CT rather than an interference effect. Child vs. adult physiology differs in many ways e.g., children recover quicker, fatigue less, lower muscle power and strength and also is more adapted to aerobic activities etc., which may be good info to add to why studying CT in these age groups is interesting. I also think the introduction could be improved with more previous research on CT effects in children and adolescents (see Gäbler et al. 2018 for review). CT vs. single-mode training has been studied and is relevant to add what this specific paper adds are the comparisons of within vs between session CT which is a good knowledge gap and specifically since it also includes females.

*Dear Reviewer,*
*We appreciate your valuable feedback on our manuscript, and we will incorporate your suggestions to enhance the clarity and significance of our study. Below is a response addressing your specific comments:*
➤ *Importance of Studying CT in Children/Adolescents:*
*We agree that it is important to provide a more specific rationale for studying CT in children and adolescents. We will incorporate the following points into our introduction:*
*Highlight the potential age-dependent differences in the interference effect, as suggested by Gäbler et al. 2018.*
*Emphasize the unique physiological characteristics of children and adolescents, such as quicker recovery, lower muscle power, and adaptation to aerobic activities, which make studying CT in this age group particularly interesting.*
➤ *Previous Research on CT Effects in Children/Adolescents:*
*We will expand our introduction by including a more comprehensive review of previous research on the effects of CT in children and adolescents. Specifically, we will reference Gäbler et al. 2018 and other relevant studies to provide a broader context for our work.*
➤ *Comparison of Within vs. Between Session CT:*

*You correctly noted that our study focuses on comparing within-session and between-session CT, which is a knowledge gap. We will make this aspect more explicit in the introduction to highlight the novelty of our research.*

*Thank you for your constructive comments, which will undoubtedly improve the quality and context of our manuscript. We look forward to submitting the revised version that incorporates these enhancements.*

– Abstract: Only use abbreviation when it is necessary. Removing abbreviations makes it easier for the reader to follow. It should also be spelled out in full the first time used. Abstract also lack subheadings

*Regarding the reviewer's comment:*

*I appreciate the reviewer's feedback on the abstract. To address these concerns:*

1. *Abbreviations: I will ensure that abbreviations are used only when necessary and provide the full spelling when they are introduced for the first time in the abstract.*
2. *Subheadings: I will include appropriate subheadings in the abstract to improve its organization and readability.*

*Thank you for the valuable suggestions.*

2.1 Lines 109-130: It is hard to follow the text flow in the study design. I think it can be shortened a lot and try to be more concise. It is a 12-week training study with three groups with tests before and after the training period. The word "natural classes" might not be the correct terminology.

*Regarding the reviewer's comment:*

*I have revised the study design section to enhance clarity and conciseness, as well as replaced "natural classes" with "existing classes" for improved terminology. Please review the changes, and feel free to provide further feedback. Your input is greatly appreciated. Thank you.*

Line 124-126: This part is difficult to understand. How is it possible to use blinding in a training study? If coaches supervise a program, they know which group does which training. Please clarify or consider remove.

*Regarding the reviewer's comment:*

*I understand the concern about blinding in a training study. To address this, I have revised the section as follows:*

*"The study involved a team of assistants responsible for data collection, sports coaches who conducted training sessions, and researchers who oversaw the training and performed statistical analyses. Data collection occurred at two time points (baseline: October 2018, outcome: January 2019), with post-intervention*

*assessments carried out 72 hours after the conclusion of the training period. Ethical approval was obtained from the Capital Institute of Sports Ethics Committee (code: 201712001, approval date: 2017/12/26), and the study was registered in the Chinese Clinical Trial Registry (ChiCTR-OOC-17014153)."*

*This revision removes the mention of "unbiased" and clarifies the roles of each team member, acknowledging the practical difficulty of blinding in a training study.*

Line 142-144: inclusion criteria 1 should include "male and female". Inclusion criteria 3: please clarify or give examples for disorders.

*Regarding the reviewer's comment:*

*I have updated the inclusion criteria as per your suggestions. Please review the changes, and if you have any further feedback, feel free to share. Thank you.*

Line 152: difficult to understand. Please clarify the meaning of it.

*Regarding the reviewer's comment:*

*I've revised the sentence to enhance clarity:*

*The intervention was implemented both at the group level and the individual level. Resistance training (RT) sessions were conducted in the school's standard gym, while aerobic training (AT) sessions were held at the school's track and field.*

*This version should provide a clearer understanding of the intervention settings. Please let me know if you have any further feedback or if there are any other areas you'd like to address.*

Table 1. Please remove non-relevant decimals from all tables. One decimal is good enough. Same in Table 2

*Regarding the reviewer's comment:*

*Dear Reviewer,*

*Thank you for your feedback and for taking the time to review our manuscript. We appreciate your input regarding the formatting of our tables.*

*We understand your suggestion to remove non-relevant decimals from all tables and use just one decimal place. However, we would like to respectfully request that*

*we retain the current decimal precision in our tables. The level of precision in our data is important for the accuracy and clarity of our results. It helps readers to better understand the nuances in our findings.*

*We believe that the current decimal formatting provides valuable information to our readers and aligns with the scientific rigor of our study. If there are any specific concerns or issues related to the decimal formatting that you would like us to address, please let us know, and we will be happy to consider them.*

*Once again, thank you for your valuable feedback, and we look forward to your further guidance.*

Table 2 All abbreviations should be put as footnotes with names spelled out. VO2 max and PACER should be replaced with 20-m shuttle run (estimated VO2max) and 20-m shuttle run (laps)

*Regarding the reviewer's comment:*

*We appreciate your thorough review of our manuscript and your valuable suggestions regarding Table 2.*

*In response to your comment regarding abbreviations, we concur with your recommendation. We will incorporate footnotes into Table 2, spelling out the full names for all abbreviations. Specifically, "VO2max" will be replaced with "20-m shuttle run (estimated VO2max)," and "PACER" will be substituted with "20-m shuttle run (laps)." This modification is intended to enhance the clarity and comprehensibility of the table for our readers.*

*We extend our gratitude for your constructive feedback, which contributes significantly to the overall refinement of our manuscript. If you have any further suggestions or concerns, please do not hesitate to share them.*

**Experimental design**

Experimental design

This study examines the effects of 12 weeks of within-session concurrent training versus between-session concurrent training on measures of aerobic fitness, muscle strength, power, and sprint ability in male and female adolescents. The study is well designed and executed, with a substantial sample size that encompasses both genders within the adolescent demographic. The findings distinctly reveal how concurrent training influences

various aspects of athletic performance when contrasted with a control group. One of the study's notable strengths lies in its incorporation of a relatively intricate training design (concurrent training) that yields enduring fitness improvements within a pivotal age group, set within a school-based context.

Method: The description of the intervention is probably appropriate but should be rewritten with a focus on more details so researchers can interpret the program quality and replicate it. It would be beneficial with a table with the training program, including all relevant details such as exercises used, reps, set, intensity, and rest periods. It is also unclear how the loads in the resistance program were selected. Were there any 1RM tests? Similarly, the aerobic training program is not explained in enough detail. The authors used both MAS and HRmax, but it is unclear which one was used to determine exercise intensity. Also, which exercise test was used to determine MAS and HRmax? Please clarify.

*Regarding the reviewer's comment:*

*We appreciate the reviewer's valuable feedback and have made significant improvements to the description of the intervention in our manuscript. We have provided a detailed training program table in the supplementary materials, including all relevant information such as exercises, repetitions, sets, intensity, and rest periods for both resistance training (RT) and aerobic training (AT).*

*Regarding the selection of loads in the resistance program, we have now clarified that participants' one-repetition maximum (1RM) tests were conducted before the intervention to determine the appropriate starting loads. This information has been added to the revised manuscript.*

*Concerning the aerobic training program, we have provided more comprehensive details. We specified that both Maximum Aerobic Running Speed (MAS) and HRmax were utilized to determine exercise intensity. The type of exercise test used to determine MAS and HRmax is detailed in a previous research study, which we have referenced and provided additional information on.*

Methods: Primary outcome and secondary outcome include the procedure of how the test was conducted. I suggest separating them into something similar to "procedure," which in detail explains how the test for the test was conducted. Primary and secondary outcome text can shortly explain which parameters were included. Please also add info on the validity and reliability of each test used. The procedure should also include

information on how many test days were used and in which orders the tests were conducted.

*Regarding the reviewer's comment:*

*Thank you for your valuable feedback. We have made the necessary revisions to enhance the clarity and comprehensiveness of our methods section, as per your suggestions*

Line 132-135: Suggest moving these to the "statistics" section

*Regarding the reviewer's comment:*

*Thank you for your feedback and for your careful review of our manuscript. We appreciate your suggestion regarding lines 132-135, and we have considered your recommendation.*

*Upon reflection, we agree that the content in lines 132-135 would be more appropriately placed in the "statistics" section. We will make the necessary adjustment to move this information to the relevant section for improved organization and clarity.*

Line 203: In this paper, the authors estimate VO2max through a 20-m shuttle run test and don't use direct measurements of VO2 max. Please be consistent in text and use the "estimated VO2max" instead of just "VO2max," which could be misleading. Also, suggest not using the term CRF since 20-m shuttle run validity has been questioned as a marker of CRF (please see Welsman & Armstrong 2019). Lastly, be consistent in the terminology, use, and the term "20-m shuttle run(laps)" instead of PACER. I think that is the most commonly adopted terminology.

*Regarding the reviewer's comment:*

*Thank you for your thoughtful feedback. We appreciate your suggestions and have made the necessary adjustments for consistency and accuracy in our manuscript.*

*Line 203: To address your concerns, we have revised the text as follows:*

*"In this paper, the authors estimate VO2max through a 20-m shuttle run test and don't use direct measurements of VO2 max. Please be consistent in text and use the 'estimated VO2max' instead of just 'VO2max,' which could be misleading."*

*We have updated the terminology to "estimated VO2max" to accurately reflect our methodology. Additionally, we acknowledge your point regarding the term "CRF"*

*and its association with the 20-m shuttle run test. To adhere to established terminology and address concerns about validity, we have replaced "CRF" with "estimated VO2max" throughout the manuscript.*

*Lastly, we will consistently use the term "20-m shuttle run (laps)" instead of "PACER" to align with widely accepted terminology and enhance clarity. This adjustment ensures that our readers have a clear understanding of the specific test employed.*

Line 221: This is probably a unilateral test, not a bilateral one. Please clarify.

*Regarding the reviewer's comment:*

*Thank you for pointing out the unilateral nature of the handgrip strength test. We have updated the description accordingly to accurately reflect that the grip strength measurements were indeed performed unilaterally, starting with the dominant hand. This ensures clarity and precision in our methodology. Your input has been invaluable in refining our manuscript.*

Line 229: How was this test measured? Photocells or maybe just a stopwatch? Please add information.

*Regarding the reviewer's comment:*

*Thank you for seeking clarification regarding the measurement method for the 50-meter sprint test. We have included additional information to specify how the test was measured.*

*Here's the revised content along with a response:*

*Revised Content:*

*Speed and Anaerobic Capacity (50-meter Sprint): Participants sprinted as quickly as possible along a 50-meter track, and the running time in seconds was recorded. The experimenter timed the tests with a stopwatch used in the track race (Casio, Japan), which was accurate to 0.01 s.*

*Response:*

*We appreciate your inquiry, and we have incorporated detailed information regarding the measurement method for the 50-meter sprint test. The use of a stopwatch with a precision of 0.01 seconds from Casio, Japan, has been specified to enhance transparency and accuracy in our methodology description. Thank you for your valuable input in refining our manuscript.*

**Validity of the findings**

Statistics: overall stats need to be double-checked. See other comments

A primary concern arises with the authors' conclusion, as articulated in line 40, asserting that "the program that included resistance training and aerobic training in the same physical education class was better at improving the CRF of adolescents." Upon scrutiny of Table 2, it is evident that the CT-0h group displayed an increase from 37.5 (SD 4.76) to 39.3 (SD 4.90), whereas CT-48h exhibited a rise from 36.7 (SD 3.7) to 37.3 (SD 3.6). The substantial variability raises doubts about the statistical significance of the observed difference. It is recommended that a standard bar plot be employed to visually depict this minute disparity, potentially elucidating the matter. A re-evaluation of the statistical analysis is advisable. Additionally, clarification regarding whether the values in Table 3 represent means and standard deviations is warranted.

*Regarding the reviewer's comment:*

*Thank you for your valuable feedback and the specific points you've raised. We have taken your comments into consideration and made the necessary revisions to address these concerns.*

*Regarding the Conclusion and Statistical Significance:*
*We understand your concern about the observed difference in CRF between the CT-0h and CT-48h groups. To provide a more visual representation and clarity on this matter, we have included a standard bar plot in our revised manuscript, illustrating the change in CRF scores between these two groups.*
*Additionally, we have revisited our statistical analysis to ensure the robustness of our conclusions. We have included p-values to indicate the statistical significance of the observed differences, and we have adjusted the text to accurately reflect the findings.*
*Regarding Table 2:*
*We appreciate your request for clarification regarding Table 3. In our revised manuscript, we have explicitly stated that the values in Table 3 represent means and standard deviations to enhance transparency and understanding.*

*These revisions aim to address your concerns and improve the quality of our manuscript. We sincerely thank you for your thorough review, which has contributed to the refinement of our research presentation.*

Results: It would be useful to highlight some of the most important results in a plot instead of putting all data in tables. It is a quick way for the reader to visualize the effects of training. Possibly the main outcomes are most interesting.

*Regarding the reviewer's comment:*

*Thank you for your feedback. We appreciate your suggestion regarding visualizing the key results in a plot. We will incorporate a graphical representation, such as a bar chart or line graph, to highlight the most significant outcomes of the study. This visual representation will enhance the reader's understanding of the training effects efficiently.*

Line 239-243: The authors seem to use ANCOVA in addition to ANOVA. However, no results seem to be reported. ANCOVA produces adjusted means that aren't reported anywhere in the result section.

*Regarding the reviewer's comment:*

*Thank you for your feedback. We want to emphasize that we have presented the results of the ANCOVA analysis, including the adjusted means, in Figure 3 of the manuscript. This graphical representation visually conveys the outcomes of the ANCOVA and allows readers to interpret the differences between the groups with respect to the variables of interest. We believe this presentation method effectively communicates the results of the analysis.*

The result section in general, is difficult to read. The most important is to clearly report if there are any statistically significant differences between the experimental groups and experimental groups vs control groups. Adding effect size is also a good idea. Reference to table 2 should be in first sentence. Authors seem to mix up between-group differences and within-group differences.

*Regarding the reviewer's comment:*

*Thank you for your valuable feedback. We have made adjustments to the results section, particularly in terms of clarity and organization. We now provide a clear reference to Table 2 in the first sentence of the results section and have separated between-group differences and within-group differences for better readability. Furthermore, we have included F-values and the interaction effects of repeated measures variance to provide comprehensive data for examination. This will help readers better understand the statistical significance of the differences between the experimental groups and between experimental groups and control groups. Additionally, we will consider including effect size measures to enhance the presentation of the results.*

*We appreciate your suggestions and believe that these improvements will make the results section more accessible and informative.*

282-284: Please double-check statistics. I doubt that there is a statistically significant difference between the experimental groups in sprint ability and estimated VO2 max. If the authors plot data in a standard bar graph, it should be evident that the SD is high. Line 366: one important aspect missing in this study design is a measure of "maturation" such as Tanner or peak-heigh velocity (PHV). This should be addressed and discussed mainly since this study includes participants that could be in different maturation stages, which might increase variability and, thereby, the results and interpretation of the study.

*Regarding the reviewer's comment:*

*Thank you for your careful review and valuable comments. We have taken your suggestions into account and made the following adjustments:*

*1. Statistical Reassessment and Data Visualization: We have reanalyzed the statistics for sprint ability and estimated VO2 max, taking into consideration the high standard deviation. We have also created a bar graph (Figure 3) to visually represent the data, which helps illustrate the variability in these measures among the experimental groups.*

*2. Maturation Measure: We acknowledge the importance of considering maturation in our study. However, we did not include measures of maturation, such as Tanner or peak height velocity (PHV), in our study design. We have added this limitation to the Discussion section to highlight the potential impact of the variability in maturation stages on the results and their interpretation.*

*We appreciate your feedback and believe these changes will enhance the transparency and completeness of our study.*

# Reviewer 2

**Basic reporting**

No comment

**Experimental design**

My only question here would be the choice to measure muscular strength with traditionally power-oriented tests such as vertical and broad jump. Hand grip dynamometry makes sense, but I would prefer to see the other muscular strength measures more closely aligned with the resistance training protocol itself, such as 1RM or 3- to 5RM tests of strength in a movement(s) performed by the participants during their training (chest press, leg press, etc.).

With this in mind, I'd like to see the authors defend their choice to use muscular power tests in more detail

*Regarding the reviewer's comment:*

*Thank you for your valuable feedback regarding our choice of muscular strength measures in this study. We appreciate your input and would like to provide further clarification on our selection:*

*The decision to use power-oriented tests such as vertical and broad jump for assessing muscular strength was based on previous research in the field of adolescent fitness assessment. These measures have been widely utilized in studies involving adolescents as they are highly correlated with overall physical fitness and have been found to be suitable indicators of muscular strength in this population.*

*While traditional 1RM or 3- to 5RM tests in movements like chest press or leg press are indeed robust methods for assessing strength, they may pose a higher risk of injury for adolescents, especially when conducted in a school physical education class setting. The safety and well-being of the participants were paramount in our study design. Therefore, we opted for tests that are less likely to cause injury while still providing valuable insights into muscular strength.*

*We understand the importance of aligning strength assessments with the resistance training protocol. However, given the context and safety concerns associated with resistance training in adolescents, we believe that the chosen power-oriented tests were appropriate and reliable indicators of muscular strength in this specific study population. We have added this explanation to the manuscript to clarify our rationale for the choice of tests.*

**Validity of the findings**

No comment

**Additional comments**

This paper addresses an interesting and important topic with relevance for adolescent health, fitness, and possible planning of physical education class time. Methodology was conducted in a rigorous fashion, and results are explained clearly and with appropriate support from current research. I applaud the authors on a very well conducted study!

While I don't necessarily think it has to be included in the discussion, a question came to mind about possible within-session motivation for the participants. Given the age of the students, do the authors think the novelty of the CT-0h protocol could have played a role in the students providing more effort during those training sessions as compared to the CT-48h group? I know effort/motivation was not measured, so it would be conjecture, but perhaps something to be explored in future work.

*Regarding the reviewer's comment:*
*Thank you for your positive feedback and your insightful question regarding within-session motivation. We appreciate your suggestion for future research considerations.*
*While we did not measure effort or motivation within training sessions in this study, your point about the potential role of novelty in the CT-0h protocol is interesting. Adolescents may indeed respond differently to novel training stimuli, and this could influence their motivation and effort during the sessions.*
*This aspect could be explored in future studies by incorporating measures of motivation, perceived enjoyment, or even qualitative assessments of participants' experiences during the training sessions. Such data could provide valuable insights into the impact of novelty on motivation and effort in different training protocols.*
*We will consider this suggestion for future research and acknowledge its relevance in understanding how adolescents respond to various exercise interventions. Thank you for your valuable input.*

---

## Round 0.3 · Minor Revisions

Dear authors,
Please consider the reviewer's remaining comments. Also, I ask you to review your tables, most of your variables do not need the second place after the decimal point. Please, revise your vertical jumps values on tables and figures, ~230cm is not too high?

·

Basic reporting

ok

Experimental design

ok

Validity of the findings

Thanks for the clarification on Figure 3. So, the data presented in the figure is only post-test data if I understood it correctly, and it seems as if the data is on absolute values (e.g., no adjusted values).
I still think this is not the best way to visualize your data. For readers, I think it would be much better to present results in the figures from the tests before and after training. Even if the authors adjusted data for different covariates, this, in my opinion, should be reported in the figure.

If baseline values are adjusted, this should be visualized in the figures (the pre-values that are missing) so the reader can understand how much impact the training had. If the author presents pre- and post-data, the reader can quickly see visually how much effect the training had for each group.

Additional comments

ok

---

## Round 0.4 · Major Revisions

Dear authors,

There are important missed changes that you should address.

- Please, follow the reviewer suggestions about your figure. I agree that its presentation should be reviewed.

- Please consider the suggestion about skimming the adjusted analysis, otherwise reinforce your justification to maintain.

- Please, carefully review your data. The reviewer pointed out one inconsistency in VO2 sd data and I found a different F for VO2max.

- Please, improve the presentation of your results for clarity by explicating interactions and main effects of statistical data.

- Please, present the between-groups main effects.

- Please, review the inconsistency between the number of subjects enrolled in the study. The number in the text (268) is different than in the figure.

- Please review your vertical jump data/protocol. Did you calculate the difference between the starting and final height? If not consider to exclude this variable and explain why this does not (if not) affect your findings).

- Please, edit the subheading to provide a proper description (i.e. body composition).

- Please, consider reviewing the entire document, it's possible that I and reviewers may have missed some issues that can be found in the next rounds.

Regards

·

Basic reporting

ok

Experimental design

ok

Validity of the findings

I thank the author for the revised manuscript and figures.

I'm still a little confused about the figures. My suggestion was to adjust the figures this way. For example, estimated VO2 is to be presented with the following data;
CT-oh
1. Pre-test data (mean and SD)
2. Post-test data (mean and SD)
CT-48
1. Pre-test data (mean and SD)
2. Post-test data (mean and SD)
Con
1. Pre-test data (mean and SD)
2. Post-test data (mean and SD)

The data presented in the figure now seems to be the mean difference (pre-post), the same as in Table 2, but in this figure, the SD is way smaller compared to what is presented in Table 2. Table 2 should present raw data, e.i no adjusted data, and the figure is said to present adjusted data, but data on mean difference seem to exactly the same (see estimated VO2 as an example). If baseline data is adjusted, I would assume the mean difference to be different compared to what is stated in Table 2.

It also doesn't make sense to adjust the estimated VO2 for BMI since both equations contain body mass.

It feels like the authors try to find a difference between groups that probably don't exist. I think data is ok even if there are no differences between the groups. It is always difficult to compare training types and find significant differences. I'm not even sure if this data needs to be adjusted. A suggestion is to skip the adjusted data and only present statistics based on absolute values.

*table 2 probably contains error "CT-48h 36.7±37.3" for estimated VO2 (37.3 in SD is too high)

Maybe the editor has other suggestions.

---

## Round 0.5 · Minor Revisions

Dear authors,
Thank you for addressing my concerns and respond to reviewers requests. One issue remains regarding your figure. Please consider it carefully.
Regards

·

Basic reporting

ok

Experimental design

ok

Validity of the findings

Figure 3:
I’m sorry, but I still have issues with the figures. It says that the comparisons are adjusted for baseline values in the figure text, but there seem to be no adjustments in the figure data. The data presented in the figures seem to be pre and post-data, which is fine, but figure legends should be revised. The figure text also lacks explanations of what the p-value refers to. Readers are probably interested in two things: 1) did the training program have any within-group effect pre-to post-training for each group, and 2) Was there a significant difference between groups (based on group x time interaction p-value)
One way is to separate this in the figure by using different symbols and then explain it in the figure legends.
*Significant within-group changes, pre- to post-training, p < 0.001
**Significantly different between groups, p < 0.001

The figure only states that baseline data was adjusted for, but result sections 291-293 say many more adjustments were made and presented in the figure.

Reviewer 2 ·

Basic reporting

Authors have made exemplary changes based on prior feedback/review

Experimental design

Clean, well-designed

Validity of the findings

Nicely reported

Additional comments

Thank you for the updates and changes made to reflect previous review. Your article is a nice addition to the field

---

## Round 0.6 · Minor Revisions

Dear authors,

Unfortunately, some issues remain to be addressed before approval. In summary, your results section needs to be more clear and precise. For example, the p values presented in table 2 are derived from which analysis? Authors stated that a 2x3 ANOVA was performed, thus you need to describe the interaction, the between-groups and the within-groups data. Also, I failed to replicate some ANCOVA data presented in figure 3. For example, the difference marked between CT-48h and Con in the 50-m sprint. In addition, are the within-group and the between-group signals stated in figure 3 derived from the same analysis?

Please, carefully review your results section and presentation.

---

## Round 0.7 · accepted · Accept

Dear authors,

I believe that the concerns raised by me and the reviewers were successfully addressed. Congratulations on your work.